# Comparative Analysis of Three Facial Scanners for Creating Digital Twins by Focusing on the Difference in Scanning Method

**DOI:** 10.3390/bioengineering10050545

**Published:** 2023-04-29

**Authors:** Ran-Yeong Cho, Soo-Hwan Byun, Sang-Min Yi, Hee-Ju Ahn, Yoo-Sung Nam, In-Young Park, Sung-Woon On, Jong-Cheol Kim, Byoung-Eun Yang

**Affiliations:** 1Department of Oral and Maxillofacial Surgery, Hallym University Sacred Heart Hospital, Anyang 14066, Republic of Korea; ran25021398@hallym.or.kr (R.-Y.C.); purheit@hallym.or.kr (S.-H.B.); queen21c@hallym.or.kr (S.-M.Y.); dave135@hallym.or.kr (H.-J.A.); handsomeu@hallym.or.kr (Y.-S.N.); ddskjc@hanmail.net (J.-C.K.); 2Graduate School of Clinical Dentistry, Hallym University, Chuncheon 24252, Republic of Korea; denti2875@hallym.or.kr (I.-Y.P.); drummer0908@hallym.or.kr (S.-W.O.); 3Institute of Clinical Dentistry, Hallym University, Chuncheon 24252, Republic of Korea; 4Division of Oral and Maxillofacial Surgery, Department of Dentistry, Hallym University Dongtan Sacred Heart Hospital, Hwaseong 18450, Republic of Korea; 5Mir Dental Hospital, Daegu 41940, Republic of Korea

**Keywords:** facial scanner, digital twin, scanning method, three-dimensional(3D), soft tissue imaging, medical images

## Abstract

Background: Multi-dimensional facial imaging is increasingly used in hospital clinics. A digital twin of the face can be created by reconstructing three-dimensional (3D) facial images using facial scanners. Therefore, the reliability, strengths, and weaknesses of scanners should be investigated and approved; Methods: Images obtained from three facial scanners (RayFace, MegaGen, and Artec Eva) were compared with cone-beam computed tomography images as the standard. Surface discrepancies were measured and analyzed at 14 specific reference points; Results: All scanners used in this study achieved acceptable results, although only scanner 3 obtained preferable results. Each scanner exhibited weak and strong points because of differences in the scanning methods. Scanner 2 exhibited the best result on the left endocanthion; scanner 1 achieved the best result on the left exocanthion and left alare; and scanner 3 achieved the best result on the left exocanthion (both cheeks); Conclusions: These comparative analysis data can be used when creating digital twins through segmentation, selecting and merging data, or developing a new scanner to overcome all shortcomings.

## 1. Introduction

Facial imaging is widely used for medical images, including various clinical purposes, such as orthodontics [1] and maxillofacial prosthodontics [2], as well as monitoring the growth and changes in soft tissue, planning surgical treatment, evaluation of postoperative results, and fabrication of facial prostheses [3,4]. While traditional two-dimensional (2D) photography has some limitations, [3] three-dimensional (3D) facial imaging can provide valuable facial information for the 3D reconstruction of patients. Virtual patients developed using 3D data are called “digital twins” [5], which can aid medical staff in diagnosis, treatment planning, explaining the treatment plan or results to patients, or surgery. Digital twins can also be used for educational purposes. Similarly, facial imaging is a significant factor in the medical area.

Volumetric or surface imaging methods are types of 3D facial imaging methods. Cone-beam computed tomography (CBCT) imaging is used as a volumetric method. This method includes internal anatomy and external imaging. However, this method requires qualified operating staff and is expensive. Moreover, artifacts can occur because of metallic materials [6]. Furthermore, exposure to ionizing radiation should be considered.

Surface imaging is an optical imaging technique. In these methods, visible light is used to capture images of only external structures. Various facial scanning devices are commercially available and widely used in dental clinics. These devices are low-cost and eliminate radiation exposure risks for patients. However, although artifacts from metallic materials do not occur, these methods have other limitations. Since these devices need to produce accurate or reproducible results with excellent soft tissue outcomes for orthodontic or maxillofacial operative planning tools [7,8], validation should be performed before releasing them to the market.

All devices are verified for comprehensive accuracy before commercialization. However, facial scanners have specific strengths and weaknesses in various facial regions. Thus, a single scanner can yield more accurate results at a specific point on the face. Therefore, verifying the points such that the most accurate soft tissue image can be fabricated by merging data from each scanner is critical.

Facial scanners can be classified in several ways. Several technologies, such as laser scanning [9], structured light pattern methods [10], and digital stereophotogrammetry [11], have been devised. Studies have validated scanners by focusing on differences in imaging technologies [12,13]. Differences can result from using different hardware, such as handheld and nonportable types. Studies have focused on hardware types [14]. However, studies that compare scanners with three different hardware types, focusing on the results at each point of the face, have yet to be published. Therefore, we judged that paying attention to these differences is necessary for both users and manufacturers. This study compared three facial scanners based on their scanning method. Our current and future research aims to develop a better facial scanner and create the perfect digital twin.

## 2. Materials and Methods

### 2.1. Patient Preparation

The procedures of this retrospective study were conducted in accordance with the World Medical Association Declaration of Helsinki and approved by the Hallym University Sacred Heart Hospital’s Institutional Review Board (IRB No. 2022-07-016).

Thirty patients were selected to participate in the study who visited the university hospital and underwent CBCT and facial scans for various treatment purposes, including prosthodontics, orthodontics, and orthognathic surgery. For the treatment of our patients, we superimpose CBCT data of the facial soft tissues with 3D data obtained using a facial scanner. Of these patients, 30 patients were selected for this study. Patients were divided into three groups based on the type of facial scanner used to obtain the three-dimensional facial images. Each scanner group included ten different patients. Patients ranged in age from 19 to 75 years and had completed facial bone growth. Patients had already provided informed consent in relation to treatment but not in relation to the study itself. Informed consent was waived by the IRB due to the retrospective nature of the study. Image data of the faces of the patients were discarded from the study dataset immediately after obtaining numerical values.

### 2.2. Facial Scanners and Scanning Procedure

Three scanners with distinct capture hardware were used. All scans were performed by the same operator (R. Y. Cho). Patients were asked to remove their facial accessories, such as glasses and earrings, and pull back the hair covering their forehead. Male participants were requested to shave to avoid minor errors in scanning. Patients were instructed to gently close their eyes and mouth and avoid facial expressions or movement during capture. Room conditions, including lighting, were controlled to imitate patients’ daily routines.

Scanner 1 involves simultaneous capture of the material. Scanner 1 is a RayFACE scanner (Ray Co., Ltd., Hwaseong, Republic of Korea), which is nonportable and captures images from three spots, namely from the left, right, and mid sides of the face horizontally. Photographs were captured in 0.5 s. Images were captured according to instructions in the manual of the manufacturer. Patients were seated upright in front of the device and looking straight ahead. The vertical height of the chair was set such that the lens fitted the level of the eyes of the patients. Capturing guidelines on the program screen revealed the recommended location and size of the ears and mouth of the patients. The operator gently moved the head and torso of the patients to fit the guidelines and horizontally aligned the Frankfort line to the floor. After capturing a photograph, data were reconstructed through the 3D rendering program RayFACE.

Scanner 2 includes material capture with rotating hardware. Scanner 2 uses a porTable 3D scanner, Intel^®^ RealSense™ Depth Camera SR305 (Intel Corp., Santa Clara, CA, USA), and a MegaGen Face scanner (MegaGen Implant Co., Ltd., Daegu, Republic of Korea). It was produced as nonportable and self-capturing hardware over the depth camera. This machine is a standing type with a camera inserted into an arm, which rotates 90° counterclockwise—180° clockwise—90° counterclockwise over the face of the patient. This device was designed to capture images of standing subjects; therefore, the patients were asked to stand in the right place and face the camera. The vertical level of the camera was adjusted according to the height of the subjects. Users could set the capture time, and the recommended rotation time by the manufacturing company was 11 s. Photograph capture required approximately 13 s, including a brief pause when the machine turned in the opposite direction. After capturing a photograph, data were reconstructed using the 3D program Dt3Dscan (MegaGen Implant Co., Ltd., Daegu, Republic of Korea).

Scanner 3 involves material capture by a moving operator. Scanner 3 is a handheld scanner, Artec Eva (Artec Group Inc., Luxembourg, Luxembourg), that is handled by an operator and requires 15–25 s depending on the operator’s skill. The patients were asked to sit on a chair with their heads slightly extended to the back side as instructed by the guidelines of the device. When capturing the image, the operator moves from the right ear to the left ear of the subject once in a direction parallel to the Frankfurt Horizontal plane. The operator then moves again to the right ear of the subject, capturing the lower side of the mandible. The capture program offered by the manufacturing company reveals the missing parts of the picture so that the operator can move around the subject, check the monitor, and fill in the missing parts. After capturing a photograph, data were reconstructed through the 3D program Artec Studio17 (Artec Group Inc., Luxembourg, Luxembourg). As recommended by the manufacturing company, the image was trimmed by an automatic algorithm available in the software, in the order of “eraser”, the menu where the operator can manually erase the unnecessary part, and then “global registration” and “fast fusion” were performed.

Table 1 summarizes the characteristics of the three scanners. Each set of scan data was saved as a stereolithography (STL) file and a wavefront object (OBJ) with texture exported as a joint photographic experts group file interchange format (JPG). Figure 1 displays images from three scanners captured by an operator under identical conditions.

### 2.3. CBCT

CBCT (Alphard 3030; Asahi, Inc., Kyoto, Japan) was used as the standard image to compare scanned images because a previous study [15] proved Alphard 3030 exhibited sufficient accuracy. CBCT was performed with a field of view 200 × 200 mm, voxel size 0.39 mm, and exposure conditions 80 kVP, 5 mA, and 17 s. The subjects’ Frankfort plane was set parallel to the horizontal plane, and they were requested to close their eyes and mouth gently and to refrain from making any facial expressions. CBCT images were converted to digital imaging and communications in medicine (DICOM) format and reconstructed in 3D.

### 2.4. Superimposition and Digital Measurement

First, after CBCT taking, the image of the outer skin of each subject was extracted from the DICOM file using digitalizing software. The DICOM file was transmitted to R2GATE™ (MegaGen Implant Co., Ltd.). Since the measurements were limited to the face area, the edit range was set to the area needed for the study, and the forehead, ears, and top of the neck were removed. Data were converted into an STL file.

Second, STL files were superimposed. The first file was the skin data acquired from CBCT, and the second was the scanned facial file. The files were first imported to Geomagic Control X by 3D Systems (3D Systems, Rock Hill, SC, USA), a digital inspection software package. Skin images of CBCT were converted to reference data (RD). The scanned facial images were measured data (MD). RD and MD images were automatically superimposed by the “initial alignment” function for the approximate position, and the “best-fit alignment” function was applied for accurate superimposition [16,17].

Third, a 3D comparison was made. A 3D color map comparison tool, which produced a color map indicating areas of adequate alignment with green color and inadequate alignment as blue or red, was used [18]. The red region, which indicates that the MD is above the RD, was set as a range of positive error from +0.5 to +1 mm, and the blue region, which indicates that the MD is below the RD, was set as a range of negative error from −0.5 to −1 mm. The set value was based on previous studies. According to other studies, a 1 mm error was the expected status of the clinically “acceptable” threshold [19,20,21,22,23,24]. In this study, “acceptable” was defined as the point having a discrepancy from 1 to 0.5 mm, and “preferable” denotes that the point has a discrepancy smaller than 0.5 mm, referring to a previous study [14].

Finally, digital measurement was done using reference points. The surface discrepancy, which denotes the 3D distance between RD and MD, was measured at 14 reference points, namely the glabella, nasion, left and right exocanthion, left and right endocanthion, left and right cheek, left and right alare, pronasale, and left and right cheilion, and pogonion (Figure 2).

The points refer to previous papers [25] but have been modified for this report. Since this study focused on evaluating the accuracy of scanners in each part of the face, points were arranged such that one to four points were distributed for each specific area of the face, namely the glabella for the forehead, nasion, pronasale, and alare for the nose, exocanthion and endocanthion for the eyes, cheilion for the lips, and pogonion for the chin. Moreover, the arrangement was conducted such that the central, upper, lower, left, and right sides of the face were evenly evaluated. The point “left and right cheek” is a temporarily designated point by authors for evaluating accuracy in both cheeks and denotes the intersection point of the vertical line passing through the exocanthion and the horizontal line passing through the alare. The digital measurement was performed by using the “comparison point” tool. The range of deviation is color graded from +1 (red) to −1 mm (blue), which denotes that the value is “acceptable”, as mentioned above, and the green color range, which indicates “preferable”, was from +0.5 to −0.5 mm. Figure 3 displays an example of the color map, and all procedures were performed using the same operator (R. Y. Cho.). Figure 3 illustrates examples of the color map and point comparison for each scanner and one of the operators.

### 2.5. General Appearance

Before statistical analysis, the “general appearance” of the color map results of all subjects was evaluated. Firstly, all operators look at the color maps and mark the red or blue areas. Next, the color maps are classified by the scanner, and the common distribution of the red and blue areas is checked for each scanner. By this “general appearance”, rough features of the results from each scanner were figured out.

### 2.6. Statistical Analysis

All surface discrepancies were indicated in the reports provided by Geomagic control X. Three statistical analyses were performed in the study using the Statistical Package for Social Science (IBM, Armonk, NY, USA), with a significance level set at α = 0.05.

First, all surface discrepancies from each scanner were compared to verify whether the scanners had clinically acceptable accuracy. This phenomenon was analyzed using previously established methods [26]. Absolute values of the surface discrepancies at each point were used. The values were classified as unacceptable, acceptable, or preferable. “Unacceptable” was defined as the point having surface discrepancies greater than 1 mm [19,20,21,22,23,24], “acceptable” reveals that the point has a discrepancy from 1 to 0.5 mm, and “preferable” indicates that the point has discrepancy smaller than 0.5 mm, referring to a previous study [14]. Initially, a normality test, the Shapiro test, was performed and obtained a CTT *p*-value of 0.000, which revealed that the values did not follow a normal distribution. Thereafter, the values were compared using the Kruskal–Wallis test.

Second, for each scanner, the discrepancies of the 14 reference points were analyzed to classify areas with unacceptable, acceptable, and preferable points. The classification was the same as described previously.

Finally, the discrepancies between the three scanners were analyzed for each reference point to determine the best scanner for each face area. The absolute values of surface differences were analyzed. The mean values of the absolute values of each scanner in the 14 reference points were compared using the Kruskal–Wallis test, which was followed by the previously described normality test.

## 3. Results

### 3.1. General Appearance

Prior to statistical analysis, the common features of the color maps were analyzed by operators. As mentioned, the red region indicates that the MD is above the RD, and the blue region indicates that the MD is below the RD.

In scanner 1, on the color map, the areas between both eyes, both sides of the nose, both sides of the chin, and both temporal areas appear blue. The bottom area of both eyes, cheek, and nose, and the top area of the eyes appeared red. The areas deeper than the surrounding area appear blue, and the area protruding to the surrounding area appear red. The area was symmetrical on both sides compared with scanner 2.

In scanner 2, on the color map, both temporal areas generally appear blue, and the right side of the face also appears blue. In contrast, the left side of the face appears red. The bottom area of the nose appears red.

In scanner 3, on the color map, general areas deeper than the surrounding area appear blue, particularly on both the nose and the nasolabial fold. Otherwise, areas that protrude from the surrounding area, such as the lateral side of the zygoma top area of the eyes and the middle of the face, including the philtrum, lips, and chin, appear red. The map was horizontally symmetrical on both sides compared with scanners 1 and 2.

### 3.2. Statistical Analysis Results

Three analyses were performed to achieve the aim of the study. First, the accuracy of each scanner was analyzed. Table 2 and Figure 4 detail the mean values and standard deviations (SDs) of the absolute surface discrepancies.

All mean values were less than 1 mm, which indicated that all the scanners exhibited acceptable accuracy. In particular, the mean value of scanner 3 satisfies the condition of “preferable” accuracy because the value was less than 0.5 mm. However, as presented in Table 2, the result revealed no significant differences.

Second, the strong and weak points of each scanner were analyzed. Table 3 and Figure 5 show the mean and SD of each reference point for the three scanners.

None of the mean values exceeded + or −1 mm, which indicated that none of the results were unacceptable. In scanner 1, points 1, 2, 3, 6, 8, 9, 10, 12, 13, and 14 had preferable results. Other points had acceptable results; points 7 and 11 had positive mean values, and 4 and 5 had negative mean values. In scanner 2, points 1, 2, 3, 4, 5, 8, 9, and 13 showed preferable results. Other points had acceptable results; points 6, 10, 11, and 14 have positive mean values, and 7 and 12 have negative mean values. In scanner 3, points 1, 2, 3, 6, 7, 9, and 11 showed preferable results. Other points had acceptable results: point 14 had a positive mean value, and points 4, 5, 8, 10, 12, and 13 had a negative mean value. The results are summarized in Figure 6.

Third, the results of each area of the face were analyzed. Table 4 presents a comparison of the results of each scanner for the 14 reference points.

Points 1, 2, 3, 4, 8, 9, 12, 13, and 14 had a significance level higher than 0.05, which reveals that the three scanners had similar results. However, points 5, 6, 7, 10, and 11 had lower significance levels, and one or two scanners whose absolute mean value was closer to 0 were picked by the post hoc test, modified by the Bonferroni test. This phenomenon indicates that the scanner had better results than the others. At point 5, scanner 2 revealed better results than scanners 1 and 3. At point 6, scanners 1 and 3 achieved better results than scanner 2. At point 7, scanner 3 showed better results than scanners 1 and 2. At point 10, scanner 1 achieved better results than scanners 2 and 3. At point 11, scanner 3 revealed better results than scanners 1 and 2. Figure 7 summarizes the results shown in Table 4.

Scanner 2 showed the best results on the left side. Endocanthion and scanner 1 had the best results on the left exocanthion and left alare, and scanner 3 had the best result on the left exocanthion and both cheeks. Other points had similar results from all scanners.

## 4. Discussion

### 4.1. Significance of the Study

With advances in 3D technology, new scanning devices for clinical applications have been developed. Although not all devices have been verified for accuracy, several studies have been conducted to verify the accuracy of scanners. Gomes et al. verified the accuracy of the Artec Eva scanner [27], scanner 3 used in our study. Celakil et al. verified the accuracy of Intel RealSense D415 (Intel Corp., Santa Clara, CA, USA), a recent version of the SR305, and scanner 2 in our study [28]. Both studies were performed by comparing the distance between the reference points on the face of the face or the conventional impression cast. Some researchers have compared the accuracy of two or more scanners. Koban et al. compared two handheld scanners with conventional nonportable devices, including the Artec Eva [14].

However, to our knowledge, a comparison of three devices for which the capturing methods differ considerably is yet to be performed. This study compared three scanners, focusing on the scanning method: simultaneous capture, capture with rotating hardware, and capture by a moving operator. Many facial scanners are newly released, but scanning methods are not unified. This study is significant because we included all possible scanning methods and offered users and manufacturers select scanners, considering the scanning method.

Furthermore, although all scanners have acceptable accuracy in capturing, each scanner has some points and surfaces with adequate or lower accuracy. To the best of our knowledge, limited studies have compared the strengths and weaknesses of scanners. For example, Koban et al. compared the surface deviations of the aesthetic units of two handheld scanners [14]. Consequently, this study compared and verified the strengths and limits of each scanner. Moreover, the future direction of our research is to extract and merge parts with high accuracy from each scanner by using various scanning methods to create a perfect digital twin.

### 4.2. Comparison of Scanners

Scanner 1 is the RayFACE scanner. This machine is nonportable. This was an installation-type device with nine cameras, all lenses capturing the subject at once and then merging the images into one 3D image. The operators only guide the subject to the right location, according to the guidelines provided in the form of ear and nose positions displayed on the monitor, and click the capture button. This phenomenon increases the convenience for operators and subjects. At this stage, the device requires a shorter time (0.5 s) for capture than the other devices. This short capture time can reduce errors caused by the movement of patients. Even if an error occurs, the short time allows the operators to capture several times conveniently, select the finest image, and reconstruct it for the 3D image. However, this type of installation has certain disadvantages. First, the installation contained nine cameras on the middle, right, and left sides of the device. Thus, the device requires sufficient space for installation. Second, the device did not have cameras to capture the lower side of the face because the nine cameras were arranged horizontally. This phenomenon results in errors at the bottom of the nose or mandible. Furthermore, the location of the camera on the right and left sides was not sufficient for capturing the temporal area of the subject. These points were unlike the handheld scanner, which can capture a picture from all directions on the face of the patient.

This scanner has several advantages. The software package released by the manufacturing company has a notable function. If users capture smiling shots and other shots with specific devices, they can merge the 3D image with an intraoral scanning file. Users can obtain a virtual postoperative image of anterior prosthetics. This image allows operators and patients to predict the results of dental operations. However, this function requires complex processes, and using the process for purposes other than aesthetic prosthetics remains challenging. Ray company released a new version of its scanner. Additional cameras were added to this machine to capture the maxillary anterior teeth and the lower side of the face. Therefore, according to the manufacturing factory, some disadvantages were overcome, and more functions were added to the software. However, considerable research is required to verify the improved accuracy and functionality of the new device version. This study revealed that the distortion in the result of scanner 1 typically occurs in curved areas. Considering the general appearance from the color map and statistical analysis of surface discrepancies, in some areas more indented than the surroundings, such as the endocanthions, the surface was captured as deeper than the reference image. In contrast, the surface was captured as shallower than the reference image in some areas that protrude into the surroundings, such as the subzygomatic and subnasal areas. This distortion originates from two problems. First, the sensitivity of the depth camera should be adjusted. Second, the lack of depth cameras captured from the bottom of the face caused distortions. By contrast, the symmetric images indicate that the location of the camera and the guidance of the subjects are favorable.

Scanner 2 is from MegaGen, a prototype model not yet commercially available depth camera (Intel ^®^ RealSense ™ Depth Camera SR305) used in a hardware shell to help operators capture images quickly. This hardware eliminates the inconvenience of the handheld depth camera. First, the subject and operator do not have to move while capturing images. The operators click on one button and wait for the results. Second, the results of the captured image do not rely on the operator’s skill and always have similarly qualified results. However, the installation of the hardware is a disadvantage, and the device is significantly bulky. The device is designed to capture standing subjects; therefore, the height of the device should be high. Moreover, since the arm of the device rotates around the head of the patient, an area greater than the length of the arm is required to install the machine. Furthermore, the scanner had difficulty scanning patients taller or shorter than average. Tall patients whose eyes were located higher than the maximum height of the camera were requested to sit on a chair and the arm was adjusted. However, capturing a picture of some patients whose eyes were lower than the minimum height of the camera was impossible. Second, despite the installation type of the device, it cannot be guided to the location of the subject, unlike the first scanner. Although the software exhibits a caution message when the subject is too far from the camera, operators remain uncertain if the face of the subject is located in the appropriate position (up, down, right, or left). Furthermore, a long time is required to capture the image of the subject (approximately 13 s). Standing subjects, who cannot hold or lean on anything from the device, can move slightly, resulting in capture errors. The device should supplement a structure that can guide and fix the location of the subjects. Third, the arm of the device rotates horizontally only. This phenomenon renders capturing the lower side of the nose and mandible difficult, similar to scanner 1. Furthermore, similar to scanner 1, the rotation angle was insufficient to capture the temporal area of the subject. This machine has several drawbacks.

The software the company produced only provides the saving function; therefore, users cannot trim or edit the image with their software and extract the image to other software for editing. Moreover, the colorization of the image was not similar to the actual color and appeared unnatural. Therefore, further research should be conducted to overcome these shortcomings. This study’s results revealed that the right side of the face generally appears blue. In contrast, the left side of the face appeared red. These results appear to originate from an error in the rotating arm. The diameter or velocity around which the arm rotates is assumed to be uneven. Furthermore, the absence of guidance regarding the location of the subject may contribute to this result. The scanner should be improved in terms of arm rotation and guidance.

Scanner 3 is the Artec Eva, a handheld 3D scanner using structured light technology. This device has been widely applied in various industrial areas and for medical purposes [14]. Compared with the other two machines, this scanner is light and portable. Thus, retaining or utilizing the device becomes manageable and does not depend on time or installation location. The position of patients does not affect the scanning results compared with previous scanners. Furthermore, unlike other scanners, which cannot scan from above or below, the portable scanner can scan from any angle toward the patient. Thus, the accuracy of the scanning results of patients’ forehead or submandibular area improves.

Moreover, this scanning method and software allowed the operator to monitor the real-time scanning process while moving around patients to the appropriate location or filling in the missing part of the image. However, this versatility can be a drawback of this scanner because scanning can be inconvenient, and the result depends on the operator’s skill. A long capturing time can cause motion defects because of patient movement. The method in which the machine captures data for a short time, or moves can compensate for this drawback. Another advantage of the scanner is its software. The program aligns the image in real-time. After the scan, all single images are fused into 3D images using geometry and texture information. The user can obtain a color-texturized 3D image composed of triangles [29]. The program has various functions for editing the image, which can help operators compensate for minor errors that occur during scanning. Some examples include the following: first, the registration menu brings all registered points into a single coordinate system and finds pair matches between the points registered in scanning. Next, the fusion menu has fast, smooth, and sharp options, creating a polygonal surface model by joining all points from the scans. Finally, the post-processing menu provides filters for erasing small objects, hole filling, mesh simplification, smoothing, and others [30]. The image appears sharp and clear because of these functions, similar to the natural face. Moreover, image coloration is expressed most naturally among the three scanners. Another disadvantage of this scanner is that the flicking LED light emitting from the machine can distract patients. The manufacturing company states that the light is not harmful to the participants. In this study, considering the general appearance from the color map and statistical analysis of surface discrepancies as a whole, similar to the results of scanner 1, the curved areas were assumed to be more severely curved. The sensitivity of the depth camera should be adjusted. However, some differences were observed between scanners 1 and 3. First, the mean value of the surface discrepancies was smaller, and the red and blue colored areas were smaller than those of scanner 1. Thus, the depth camera is more delicate and accurate. Operators can proceed while checking the result of capturing improves accuracy, in contrast to scanner 1. Second, scanner 3 has a better result on the bottom areas of protruding parts, such as the subnasal, subzygomatic, and submandibular areas, because scanner 3 can capture images from all directions, in contrast to scanner 1, which can capture images only from a limited direction.

### 4.3. Limitations of the Study

This study has the following limitations. First, in addition to the scanning methods, other differences exist among scanners, such as surface imaging technology, resolution, and software programs. These differences could have been device-specific and not only due to the scanning methods. However, since the study focused on finding differences by areas within the face, other factors are unlikely to cause a significant error. Furthermore, these device-specific factors showed us other meaningful findings when comparing scanners, such as the convenience or inconvenience of each scanner. The availability of other scanners with each scanning method would have been efficient. However, if new studies are written based on the result of this study and data accumulates, a more precise result will be obtained. Second, a comparison of the three scanners was not performed on the same patient. Each patient was scanned by only one scanner. This limitation occurred because the study was retrospective and used previously captured data for various purposes. We are in the process of superimposing facial soft tissue from CBCT on 3D surface data from a facial scanner in the treatment of patients. During this process, specific points had to be calibrated. In this situation, the accuracy of the image acquisition of each facial scanner must be checked. Therefore, since the images obtained with the facial scanner were compared with the facial soft tissue obtained from the CBCT of the same patient, we decided that it was not ethical to perform three facial scans on the same patient. Third, the precision of the scanners was not measured; all data were not acquired repeatedly because this study was retrospective. With regards to the second limitation, this shortage could have been overcome if mannequins were used instead of patients. The use of mannequins in the study would also have several advantages. If unified mannequins are used in the study, the subjects have no movement, and it is easy to control and create a similar experimental environment. However, these points can also be a disadvantage. Facial scanners are used in reality, with real humans, but using mannequins is just an experiment in a laboratory. Unlike objects, humans have various appearances, and some situations cannot be controlled by them. For example, in the study, in the case of scanner 2, certain limitations of the scanner were observed depending on the height of the patients or when the patients had no place to lean on, which could cause movement. These disadvantages could never be discovered if the subject was immovable, such as a mannequin. Similarly, to learn about these advantages and disadvantages, use on real patients instead of mannequins was required.

Finally, the reference points cannot cover all areas on the faces. More points should have been placed on wide spaces such as the forehead or cheeks. However, no standardized or objective points were observed on the forehead or cheeks because there are no distinctive structures in that area in contrast to the eyes, nose, or chin. This study created a temporary point for researching the cheek points. In contrast, analysis by segmented area, not by points, such as in another study [14], would have provided superior results. However, since distinctive and standardized boundaries on the face are absent, segmenting the area would not be objective. Moreover, other studies show that measuring points have adequate reliability [25,31,32].

This study verified the strengths and drawbacks of each scanner to fabricate the most accurate soft tissue image by merging data. If an operator uses the three scanners, data can be segmented, selected, and merged according to the results of this study, as shown in Figure 6. However, if a single scanner that could overcome all the disadvantages of these scanners were released, accurate data could be obtained more conveniently, quickly, and cheaply.

## 5. Conclusions

All scanners in this study had acceptable results, although only scanner 3 had preferable results. Scanners 1, 2, and 3 had mean values of 0.5277, 0.5098, and 0.4823, respectively. Each scanner had distinct weak and strong points because of the differences in the scanning methods. In the analysis of each point, scanner 2 achieved the best result on the left endocanthion, and scanner 1 had the best result on the left exocanthion and left alare. Scanner 3 had the best result on the left exocanthion and both cheeks. Other points resulted from all scanners. These comparative analysis data can be used when creating digital twins by segmenting, selecting, and merging data or developing a new scanner to overcome all shortcomings.

## Figures and Tables

**Figure 1 bioengineering-10-00545-f001:**
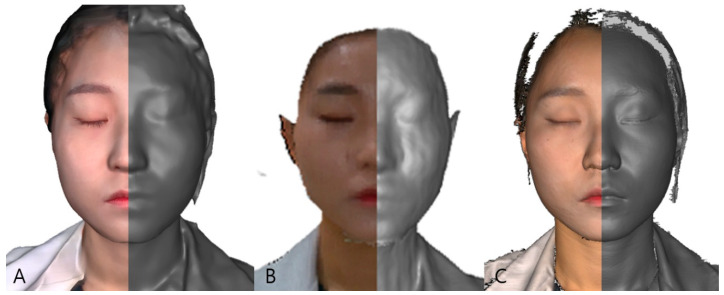
Examples of 3D scans captured by RayFACE (**A**), MegaGen scanner (**B**), and Artec Eva (**C**) of the same object. Frontal view of right halves with an untextured STL file and left halves with a textured OBJ file. (These three-dimensional images were obtained from one author for illustrative purposes.).

**Figure 2 bioengineering-10-00545-f002:**
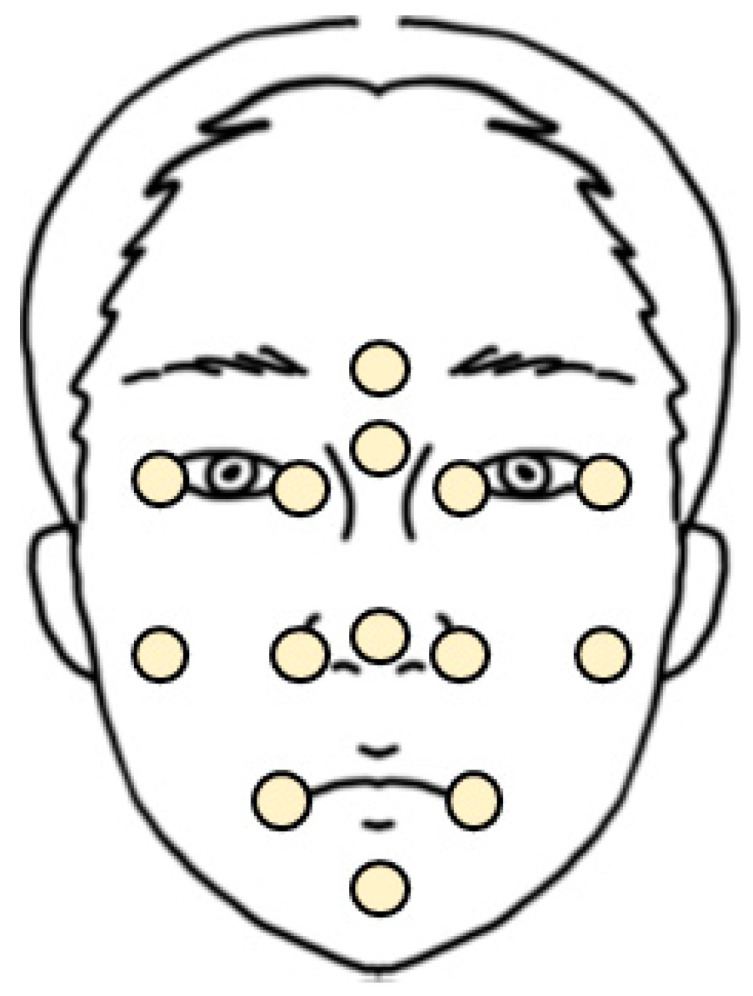
Illustration of the face and 14 reference points.

**Figure 3 bioengineering-10-00545-f003:**
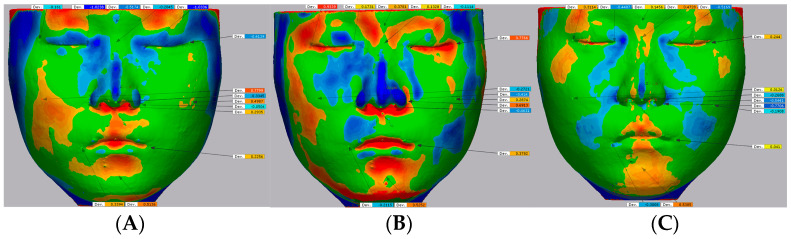
Examples of the color map and point comparison captured by RayFACE (**A**), MegaGen scanner (**B**), and Artec Eva (**C**) of the same object. The color bar on the right represents the range of deviation. (These three-dimensional images were obtained from one author for illustrative purposes.).

**Figure 4 bioengineering-10-00545-f004:**
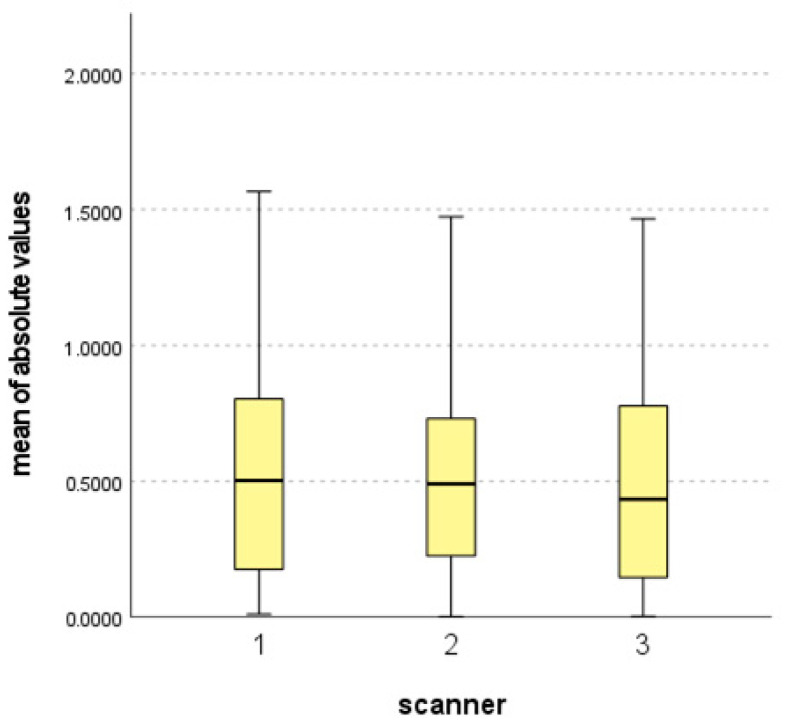
Mean, SD, minimum, and maximum of all absolute values from each scanner.

**Figure 5 bioengineering-10-00545-f005:**
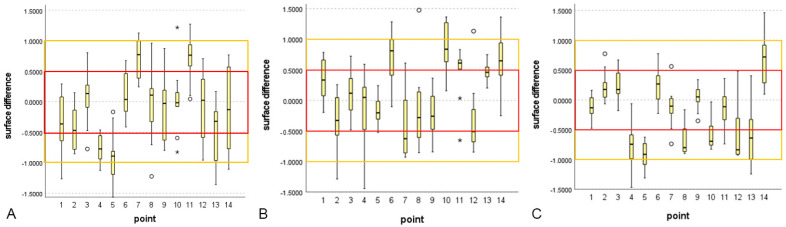
Mean, SD, minimum, and maximum of all surface discrepancies from each scanner. The orange box indicates that the surface discrepancy is less than 1 mm, and the red box indicates that the surface discrepancy is less than 0.5 mm. * and ◦ in the graph means outliers. (**A**) scanner 1, (**B**) scanner 2, and (**C**) scanner 3.

**Figure 6 bioengineering-10-00545-f006:**
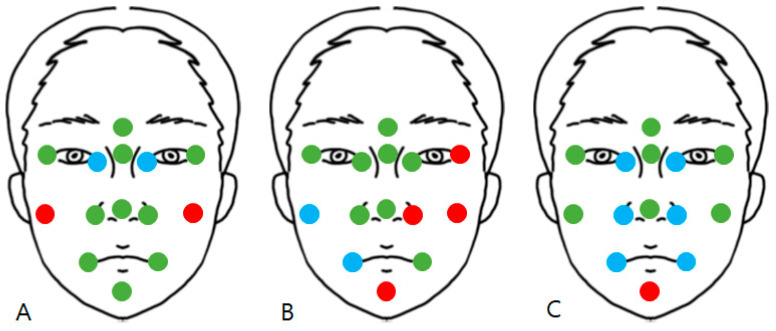
Summary of results from Table 3 and Figure 5. The green point indicates preferable results, the red points indicate acceptable positive results, and the blue points indicate acceptable negative results. (**A**) scanner 1, (**B**) scanner 2, and (**C**) scanner 3.

**Figure 7 bioengineering-10-00545-f007:**
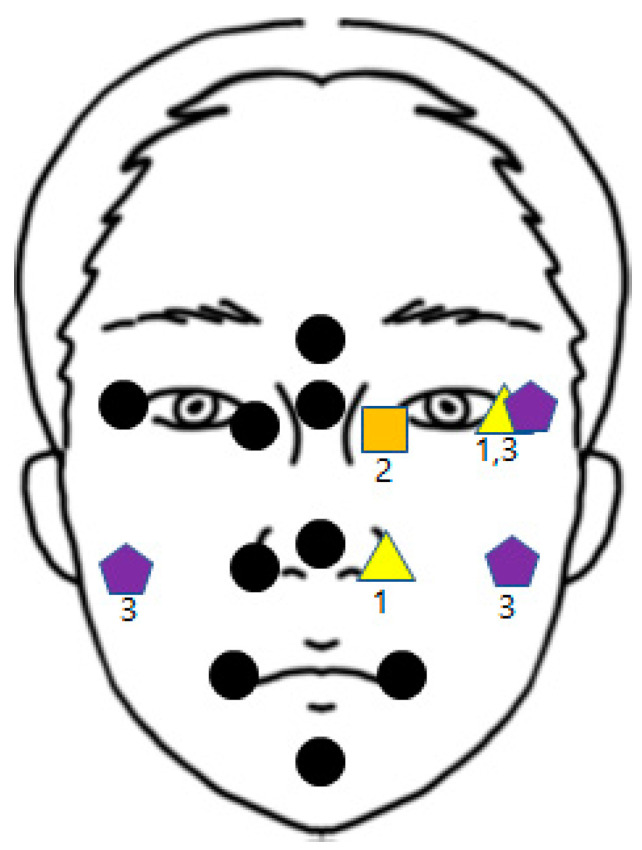
Summary of Table 4. The black circles indicate points where all scanners have similar results. The written numbers indicate that a specific scanner performs better in this area than others.

**Table 1 bioengineering-10-00545-t001:** Comparison of three facial scanners.

	Scanner 1	Scanner 2	Scanner 3
Photo	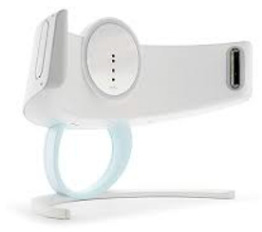	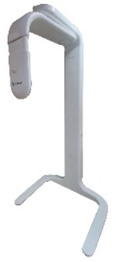	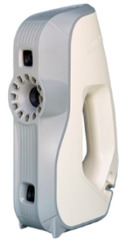
Product	RayFACE (RFS100)	MegaGen Face scanner Intel^®^ RealSense™ Depth Camera SR305 *	Artec Eva
Company	Ray Co., Ltd. (Hwaseong, Republic of Korea)	MegaGen Implant Co., Ltd. (Daegu, Republic of Korea) Intel^®^ *	Artec 3D (Luxembourg, Luxembourg)
Hardware	Horizontally curvilinear shaped, cameras at the middle, right and left side	Standing type with a camera inserted in the arm, which rotates horizontally over the head of the patients	Handheld scanner with flash bulb
Portability	Not feasible	Not feasible	Portable device
Surface imaging technology	Dual structured light	Coded light (rolling shutter)	Structured light
Lens	Nine cameras (3 cameras from 3 angles, 2 RGB cameras, and one depth camera at each angle)	One camera (depth camera with 2MP RGB sensor)	Three cameras (top, middle, bottom of the machine, middle camera surrounded by flash LEDs to get texture information)
Resolution	2 Mega Pixel	Depth 640 × 480 pixel RGB 1920 × 1080 (30 fps)	0.2 mm
Acquisition time	0.5 s	14 s	15–25 s **
Dimensions (H × W × Dia.)	813 mm × 500 mm × 550 mm	2013 mm × 1000 mm × 925 mm	261.5 mm × 158.2 mm × 63.7 mm
Field-of-view	550 mm × 310 mm	Depth: 69° ± 3° × 54° ± 2° RGB Sensor: 68° × 41.5° (±2°)	400–1000 mm × 400–1000 mm
Weight	12 kg/26.5 lbs	Not measured	0.9 kg
Output formats	STL, OBJ, Polygon file format	OBJ	All formats (STL, OBJ, PLY)
Power	Power cord	Power cord	Power cord
Processing Software	RAYFace	Dt3DScan	Artec Studio 17

* Portable 3D scanner produced by Intel^®^ with hardware shell produced by MegaGen Implant Co., Ltd. ** Depends on the operator.

**Table 2 bioengineering-10-00545-t002:** Mean (mm), SD (mm), and *p*-value of all absolute values from each scanner.

Scanner	Mean (SD)	*p*-Value
1	0.5277 (0.3717)	0.559
2	0.5098 (0.3484)	
3	0.4823 (0.3594)	

**Table 3 bioengineering-10-00545-t003:** Mean and SD of values of each reference point in three scanners.

Point	Mean (SD)		
	Scanner 1	Scanner 2	Scanner 3
1	−0.3253 (0.4678)	0.3392 (0.3554)	−0.1183 (0.1989)
2	−0.4054 (0.3690)	−0.3396 (0.4441)	0.2272 (0.2634)
3	0.0612 (0.4404)	0.1297 (0.3725)	0.2328 (0.2607)
4	−0.7544 (0.2196)	−0.1774 (0.6493)	−0.7864 (0.3807)
5	−0.8957 (0.4364)	−0.1862 (0.2393)	-0.9172 (0.2147)
6	0.1095 (0.3647)	0.7075 (0.4166)	0.2407 (0.3134)
7	0.6839 (0.3202)	−0.4093 (0.5304)	−0.1203 (0.3441)
8	−0.0018 (0.6498)	−0.1278 (0.6522)	−0.6653 (0.2525)
9	−0.1244 (0.5197)	−0.1986 (0.3845)	0.0479 (0.2234)
10	0.0248 (0.5453)	0.8559 (0.3859)	−0.5992 (0.2445)
11	0.7021 (0.3955)	0.4638 (0.4502)	−0.1501 (0.3230)
12	−0.1282 (0.5815)	−0.2921 (0.5779)	−0.6178 (0.4579)
13	−0.4812 (0.4805)	0.4797 (0.1629)	−0.5937 (0.4828)
14	−0.1653 (0.7032)	0.6639 (0.4426)	0.6976 (0.5597)

**Table 4 bioengineering-10-00545-t004:** Comparison of mean of absolute values of each scanner in 14 reference points.

Point	Significance Level	Post Hoc Test *	Comparison by Scanners **
1	0.660		1 = 2 = 3
2	0.171		1 = 2 = 3
3	0.951		1 = 2 = 3
4	0.070		1 = 2 = 3
5	0.000	S2–S1: 0.003, S2–S3: 0.001, S1–S3: 1.000	2 > 1 = 3
6	0.013	S3–S1: 1.000, S3–S2: 0.029, S1–S2: 0.036	1 = 3 > 2
7	0.018	S3–S2: 0.170, S3–S1: 0.16, S2–S1: 1.000	3 > 1 = 2
8	0.229		1 = 2 = 3
9	0.107		1 = 2 = 3
10	0.015	S1–S3: 0.586, S1–S2: 0.011, S3–S2: 0.329	1 > 2 = 3
11	0.015	S3–S2: 0.105, S3–S1: 0.017, S2–S1: 1.000	3 > 1 = 2
12	0.259		1 = 2 = 3
13	0.359		1 = 2 = 3
14	0.792		1 = 2 = 3

* Significance levels from the post hoc test. S1: scanner 1, S2: scanner 2, S3: scanner 3. ** “=” means the scanners have similar results; “>” means the former scanner has a better result than the latter scanner.

## Data Availability

The datasets generated during and/or analyzed during the current study are available from the corresponding author upon reasonable request.

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
