# Peer review of "Comparative Analysis of Three Facial Scanners for Creating Digital Twins by Focusing on the Difference in Scanning Method"

_bioengineering, 2023, doi:10.3390/bioengineering10050545_

Round 1

Reviewer 1 Report

This article entitled, ‘Comparative Analysis of Three Facial Scanners for Creating Digital Twins by Focusing on the Difference in Scanning Method’ by Ran-Yeong Cho et al.

In this paper, the authors evaluated three facial scanners. The authors concluded that all scanners used in this study achieved acceptable results, although only scanner 3 obtained preferable results.

The results of the study are useful information for many people working in the medical field. However, the methods and discussion sections are inadequate for a scientific paper. 

I have the following concerns.

1. It is unclear where the benefit of using patients. Please explain why mannequins should not have been used.

2. Only one scanner was used for each scanning method to make measurements. We don't know if the results of this study depend on the scanning method or if they are device-specific.

3. The authors wrote a lot of product descriptions in the discussion. However, their information could not explain the results of this study.

4. As the author admits in the limitations of the study section, it is fatal to evaluate scanners by different patients in this study.

Reviewer 2 Report

The authors present a comparative analysis of 3 facial scanners based on the differences of 14 facial points. In this sense, I consider that some aspects can be improved.

 1)    I recommend an expanded literature review to strengthen this proposal. For example, papers that have used the scanners studied in this paper. I recommend emphasizing the differences and the scientific contribution of this analysis.

2)    Were images taken from different studies? It is not clear.

3)    Was the same study obtained for each patient in each scan? How many images were compared in total?

4)    I recommend adding the figures immediately after you mention each one in the text.

5)    In section 2.6 I consider it important to present the data obtained to apply the statistical analysis. Especially to reinforce, the analysis and the statements such as “The values were compared using the Kruskal–Wallis test and the normality test, which revealed that the values did not follow a normal distribution”

6)    In section 3, the following statement is unclear “On the color map, the areas between both eyes, both sides of the nose, both sides of the chin, and both temporal areas appear blue. The bottom area of both eyes, cheek, and nose, and the top area of the eyes appeared red. The areas deeper than the surrounding area appear blue, and the area protruding to the surrounding area appear red. The area was symmetrical on both sides compared with scanner 2.” Was this found in the images of the 30 patients?

7)    I recommend not numbering subsections such as 3.1.1, 3.1.2, and 3.1.3. The information can be part of the general aspects section without being divided into a subsection. Even subsections 3.2.1, 3.2.2, and 3.2.3 can also be unnumbered.

8)    I suggest that the information in subsection 4.1 be included in the introduction section or as part of a new state-of-the-art review section.

9)    According to the results obtained from the 3 scanners, I consider it essential that the authors emphasize if analyzed scanners are suitable for the generation of digital twins.

10)  I consider that the research proposed by the authors in the paper is interesting and has the potential to be published if the authors point out their contribution and originality. Although, they indicated that no studies compare results focused on each point of the face. It is essential to emphasize the scientific contribution and the purpose. Some questions are important to answer, such as: Who will benefit from scanner benchmarking?

11)  In addition, aspects of document structure can be improved, especially the numbering of subsections. They include very short subsections that do not need to be divided into other subsections.

Round 2

Reviewer 1 Report

The authors have addressed all my concerns. The paper has been significantly improved. The revised manuscript is accepted in its present form.

Reviewer 2 Report

The authors have addressed all my comments. The content of this paper has been improved significantly. This paper can be accepted in its present form.